
# Calibration of Absorbing Boundary Layers for Geoacoustic Wave Modeling in Pseudo-Spectral Time-Domain Methods

Carlos Spa[1], Otilio Rojas[1,2], and Josep de la Puente[1]

[1]Barcelona Supercomputing Center. Barcelona, Spain.
[2]Department of Computer Science of Universidad Central de Venezuela, Caracas, Venezuela.

**Correspondence:** Carlos Spa (cspacarv@bsc.es)

**Abstract.** This paper develops a calibration methodology of the artificial absorbing techniques typically used by Fourier pseudo-spectral time-domain (PSTD) methods for geoacoustic wave simulations. Specifically, we consider the damped wave equation (DWE) that results from adding a dissipation term to the original wave equation, the sponge boundary layers (SBL) that apply an exponentially decaying factor directly to the wavefields, and finally, a classical split formulation of the Perfectly Matched Layers (PML). These three techniques belong to the same family of absorbing boundary layers (ABL), where outgoing waves and edge reflections are progressively damped across a grid zone of $N_{ABL}$ consecutive layers. The ABLs used are compatible with a pure Fourier formulation of PSTD. We first characterize the three ABL with respect to multiple sets of $N_{ABL}$ and their respective absorption parameter for homogeneous media. Next, we validate our findings in heterogeneous media of increasing complexity, starting with a layered medium and finishing with the SEG/EAGE 3-D Salt model. Finally, we algorithmically compare the three PSTD-ABL methods in terms of memory demands and computational cost. An interesting result is that PML, despite outperforming the absorption of the other two ABLs for a given $N_{ABL}$ value, requires approximately double the computational time. The parameter configurations reported in this article, can be readily used for PSTD simulations in other test cases, and the ABL calibration methodology may be applied to other wave propagation schemes.

## 1 Introduction

The Fourier Pseudo-Spectral Time-Domain (PSTD) method has been applied to wave propagation problems in, e.g., electromagnetism Filoux et al. (2008), photonics Pernice (2008); Li et al. (2000), room acoustics Spa et al. (2011), or outdoor acoustics Hornikx et al. (2010), among others. It is based upon replacing the spatial derivatives with their equivalent in the Fourier domain. If computed on Cartesian grids, the spatial accuracy order of PSTD is proportional to the amount of grid nodes in each direction and wavefields can be accurately modelled with as few





as two points per minimum wavelength, i.e. only limited by the Nyquist-Shannon theorem. The frugal require-
ments of the method made it popular in early applications to seismic modeling in the 1980's, given the limited
computer memories available at the time. For example, we find Kosloff and Baysal (1982); Fornberg (1987, 1988);
Etgen and Dellinger (1989); Daudt et al. (1989). Recent applications have focused on complex Earth models and
parallel implementations, see for instance Klin et al. (2010); Peng and Cheng (2016); Xie et al. (2016, 2018). PSTD
applications in geophysics are typically defined on unbounded domains or half-spaces, thus requiring effective
numerical methods to avoid reflections from the computational boundaries of the domain under study. This is a
restriction that can be found in all other numerical methods for wave propagation, but is more relevant for PSTD
methods. The main reason is that Fourier transforms assume periodicity of wavefields at domain boundaries.
A decaying value of the variables towards zero at the boundaries is a possible solution that ensures periodicity.
However, if there are imperfections in such decay, strong numerical errors related with the Gibbs phenomenon
Fornberg (1998); Canuto et al. (1988) can manifest. Periodicity at artificial boundaries can be achieved, for ex-
ample, by means of absorbing boundary layers (ABL), where outgoing waves and edge reflections are gradually
attenuated along several grid layers until reaching the domain's boundary. Such ABL are characerized by the bal-
ance between the number of absorbing layers used before each boundary and the parameters chosen to control
the rate of the absorption, i.e. how abruptly the absorption increases at each layer of the ABL. Too strong an
aborption profile will result in reflected energy within the absorbing layers and too weak and absorption profile
will result in high-amplitude waves reaching the boundary and thus reflecting back into the domain.

It is worth considering that ABL are not the only techniques used to absorb waves in numerical simulations. For
example, Reynolds- or Higdon-type absorbing boundary conditions (ABC) Reynolds (1978); Higdon (1986, 1987)
impose values on the variables directly at the boundary, usually splitting the wave equation into one-sided ver-
sions locally. Nevertheless, such ABCs have not been adopted in PSTD methods, to our knowledge, and thus are
not part of this work.

The first ABL technique that we will consider in the present work is the damped wave equation (DWE) Israeli
and Orszag (1981) that follows a simple analytical formulation by adding a dissipative term directly to the acous-
tic wave equation. Remarkably, the physical connotation of the damping term facilitates the formal analysis of
reflection and transmission coefficients at the ABL region for acoustic waves, and also enables similar formula-
tions and analyses of DWE for alternative propagation models. Such formulations and studies were undertaken
in the pioneering work Kosloff and Kosloff (1986), that also presents an early DWE application to 2-D PSTD acous-
tic modeling. Recently, Spa et al. (2014) presented an analytical and numerical study on optimal damping pro-
files of DWE applied to PSTD acoustic wave propagation. Besides the aforementioned studies we have found no
literature analyzing DWE for PSTD. Additional studies using DWE for a variety of wave phenomena and finite
difference (FD) methods can be found in Israeli and Orszag (1981), Sochacki et al. (1987) and Bodony (2006).

The second ABL technique that will be analyzed is the sponge boundary layer (SBL) proposed in Cerjan et al.
(1985). Here, the amplitude of wavefields are progressively attenuated by directly applying to them a damping





factor of increasing value at the absorbing layers. This technique does not stem from a modified wave equation and its underlying principles are unclear. Nevertheless, Cerjan et al. (1985) provided with a recomendation re-
garding ABL size and damping factor and, due to its simplicity, SBL has been widely used for PSTD applications Reshef et al. (1988); Fornberg (1998). There exist several applications to FD schemes as well, such as Bording (2004); Dolenc (2006); Matsumoto et al. (2009). In particular, Bording (2004) proposes alternative optimal values for SBL size and damping.

The third and last ABL analyzed are the Perfectly Matched Layers (PML). PML were introduced in electromag-
netism by Bérenger (1994, 1996) and rapidly became an absorbing method of choice in this field, see e.g. Chew and Weedon (1994); Kaufmann et al. (2008); Bérenger (2015). Following its success for electromagnetism, the method was successfully adapted to seismic modelling (e.g., Chew and Liu (1996); Komatitsch and Tromp (2003); Kristek et al. (2009)). The coupling of PML to PSTD methods starts with the pioneer work by Liu Liu (1998a) to simulate acoustic wave propagation in heterogeneous media, followed by studies in similar and more general
rheologies Liu (1998b); Klin et al. (2010); Giroux (2012); Spa et al. (2014); Xie et al. (2016). The analytical, contin-uous, PML formulation results in a reflection-less interface between physical domain and ABL. However, upon discretization, the discrete damping profiles reflect energy back to the domain and, more importantly, instabil-ities arise. Therefore a problem-dependent optimization of PML parameters must be undertaken to find stable and efficient discrete implementations. In the case of FD methods, some examples include Lisitsa (2000); Ko-
matitsch and Martin (2007); Kristek et al. (2009).

In this work, we compare the characteristics of all three ABL methods mentioned above combined with PSTD schemes. In Section 2, we present the mathematical formulation of the ABL methodologies under study, in the framework of PSTD methods, as well as theoretical aspects specific to each of them. In Section 3, we perform a calibration of ABL parameters in homogeneous media, by means of analyzing the energy absorbed and the ac-
curacy of seismic experiments for a massive simulation set. In Section 4 we use results from the calibration and analyze their validity for two different heterogeneous test cases. Finally, in Section 5, we introduce an analysis re-garding the memory footprint and computational time required by each ABL technique in a realistic application. Finally, in Section 6, we present our concluding remarks and future work.

## 2 The Fourier PSTD method and ABL types

The Fourier PSTD method can be considered a particular case of finite differences (FD) on Cartesian grids where spatial derivatives are substituted with differentiation in the spectral (Fourier) domain. This means that any spa-tial derivative requires a forward and inverse Fourier transform for the direction differentiated. By multiplying the variable in the spectral domain by $(\iota k)^n$ we obtain the $n$-th derivative of the variable, with $\iota$ the imaginary unit and $k$ the wave number. In the particular case of the linear wave equation, or constant-density acoustic wave
equation, two formulations are popular. On one hand the first-order velocity-pressure formulation, also known





as Euler formulation, in absence of forcing terms, reads

$$\frac{\partial p}{\partial t} = -\rho c^2 \boldsymbol{\nabla} \cdot \mathbf{v} + s \,, \tag{1}$$

$$\rho \frac{\partial \mathbf{v}}{\partial t} = -\boldsymbol{\nabla} p \,, \tag{2}$$

where $p$ is pressure, $\mathbf{v}$ the particle velocity, $\rho$ the density (taken constant and homogeneous), $c$ the wave speed
and $s$ a known source term. On the other hand the second-order equation where the only variable is pressure,
which reads

$$\frac{\partial^2 p}{\partial t^2} = c^2 \Delta p + \frac{\partial s}{\partial t} \,, \tag{3}$$

where $\Delta$ is the Laplacian operator. The parameter $c = c(x,y,z)$ can vary spatially and the variables $p = p(x,y,z,t)$
and $\mathbf{v} = \mathbf{v}(x,y,z,t)$ can also evolve in time. The source term $s = s(x,y,z,t)$ will be omitted in the following. We
restrict our analysis to sources that are finite in space and time and differentiable.

The Euler formulation tends to be less efficient than the second-order formulation, because it requires more
spatial variables to be stored and differentiated, but is well suited to some numerical applications where first
derivatives are relevant. This is the case, for example, of the classical split PML formulation that depends on
directional derivatives of both the pressure and velocity fields. Other ABL such as DWE and SBL do not require
additional differentiation and thus can be solved directly using the second-order formulation.

In the following we will use Cartesian regular grids, where all spatial differential operators employ forward and
inverse 1-D Fast Fourier Transforms (FFT) along each Cartesian direction. We will consider constant time and
spatial sampling, $\delta_t$ and $\delta$, respectively. Hence we discretize space and time according to $(x,y,z,t) \sim (i\delta, j\delta, l\delta, n\delta_t)$
and will use the triplet $(i,j,l)$ to describe any point in the spatial grid, while using the $n$ index to describe the time
step. Under the aforementioned discretization, the Laplacian operator applied to the variable $p$ results in

$$\Delta p|_{i,j,l}^n \approx \mathcal{F}_x^{-1}\left[(\iota k_x)^2 \mathcal{F}_x[p|_{:,j,l}^n]\right] + \mathcal{F}_y^{-1}\left[(\iota k_y)^2 \mathcal{F}_y[p|_{i,:,l}^n]\right] + \mathcal{F}_z^{-1}\left[(\iota k_z)^2 \mathcal{F}_z[p|_{i,j,:}^n]\right] \quad, \tag{4}$$

where $\mathcal{F}$ and $\mathcal{F}^{-1}$ denote the 1-D discrete Fourier Transform and its inverse, respectively, and the subindex in-
dicates the direction of transformation. Furthermore $\mathbf{k} = (k_x, k_y, k_z)$ is the wavenumber vector, $\iota = \sqrt{-1}$ and the
: symbol refers to the indexes affected by the transforms. Our computational domains may be either fully un-
bounded or a half space. In the former case ABLs apply to all six faces of the domain whereas in the latter five
faces require ABLs and at the top face a free-surface condition is applied. In all examples in this work we will use
second-order explicit time stepping based upon finite-differencing the time derivatives. Higher-order in time
versions of PSTD can be found in Spa et al. (2020), which could be applied to the ABLs described here with some
modifications.

## 2.1 Generalizations of the Absorbing Boundary Layers

All ABLs considered in the following will be presented using a unified representation of the grid. We will assume
that the computational domain includes both grid points of the physical domain and grid nodes of the absorbing





layers. The grid of the physical domain has size $(N_x, N_y, N_z)$ and we will consider $N_{\mathrm{ABL}}$ nodes added at each of the six faces of the domain as absorbing boundary layers. Furthermore, an additional node at the boundary of

the computational domain is added, whose variable value is forced to zero at each time step. Figure 1 illustrates a 2D slice of such a grid. The area inside the dashed line is the physical domain and the outside grid nodes belong to the absorbing layers and boundary. In the following we will consider that the extent of source terms is confined to the physical domain. Each grid node within the absorbing layers has a characteristic distance to the physical domain named $d$ where $d_{i,j,l} = \sqrt{(d_{i,j,l}^x)^2 + (d_{i,j,l}^y)^2 + (d_{i,j,l}^z)^2}$ and $d_{i,j,l}^\beta$ is the distance in grid nodes from $(i,j,l)$ to

the closest node of the main grid in the $\beta \in \{x,y,z\}$ direction. In the Figure, the gray scale represents the value of $d$ at each point of the boundary layers. The definition of suitable absorption parameters for each ABL that depend explicitly on $d$, and become zero inside the main grid (i.e. when $d = 0$) allows all ABL formulations in the following to use a global updating scheme. In other words, the same scheme is applied equally to all grid points in the computational domain, regardless of them being part of the physical domain or the absorbing layers. There

remains a last issue in order to solve the wave equation in the computational domain from parameters of the physical domain: The velocity $c(x,y,z)$ is only defined within the physical domain. However we need to assign a velocity value to each node in the computational domain in order to solve the wave equation. We choose in the following a direct continuation strategy where all absorbing-layer nodes take their velocity value from the closest physical-domain grid-node velocity value. For a homogeneous model this result in the whole computational domain sharing the $c$ value of the physical domain.

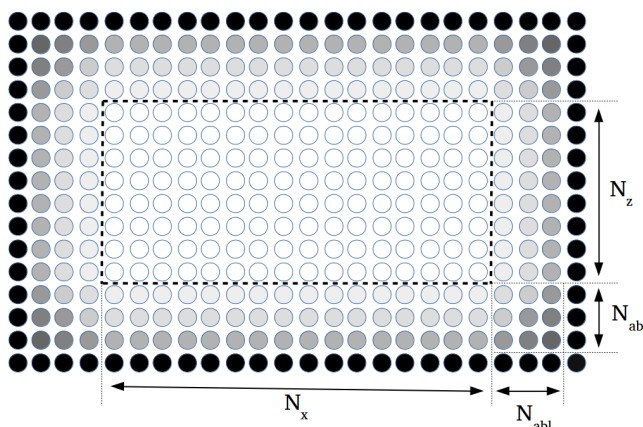

**Figure 1.** A vertical cross section of the computational mesh, along a $y$ grid plane, where $d$ is shown in a grey colored scale. Finally, black dots are nodes where pressure is forced to have a value 0.






## 2.2 The damped wave equation (DWE)

The DWE is derived from the linear wave equation (3) by adding a dissipative term that depends on the first-order temporal derivative of the acoustic pressure, and reads

$$\frac{\partial^2 p}{\partial t^2} + \sigma \frac{\partial p}{\partial t} = c^2 \Delta p, \tag{5}$$

where $\sigma = \sigma(x, y, z)$ is the coefficient of the damping term.

We use the standard second-order and central FD approximations for both temporal derivatives in (5). Furthermore, we split the discrete acoustic pressure into Cartesian projections, i.e.,

$$p|_{i,j,l}^{n+1} = p_x|_{i,j,l}^{n+1} + p_y|_{i,j,l}^{n+1} + p_z|_{i,j,l}^{n+1}, \tag{6}$$

where these acoustic projections $p_x|_{i,j,l}^{n+1}$, $p_y|_{i,j,l}^{n+1}$ and $p_z|_{i,j,l}^{n+1}$, are updated according to

$$
\begin{aligned}
\quad p_x|_{i,j,l}^{n+1} &= \frac{\sigma_{i,j,l}\delta_t - 2}{\sigma_{i,j,l}\delta_t + 2} p_x|_{i,j,l}^{n-1} + \frac{4}{\sigma_{i,j,l}\delta_t + 2} p_x|_{i,j,l}^{n} + \frac{2(c_{i,j,l}\delta_t)^2}{\sigma_{i,j,l}\delta_t + 2} \mathcal{F}_x^{-1} \left[ (\iota k_x)^2 \mathcal{F}_x \left[ p|_{:,j,l}^{n} \right] \right], \\
p_y|_{i,j,l}^{n+1} &= \frac{\sigma_{i,j,l}\delta_t - 2}{\sigma_{i,j,l}\delta_t + 2} p_y|_{i,j,l}^{n-1} + \frac{4}{\sigma_{i,j,l}\delta_t + 2} p_y|_{i,j,l}^{n} + \frac{2(c_{i,j,l}\delta_t)^2}{\sigma_{i,j,l}\delta_t + 2} \mathcal{F}_y^{-1} \left[ (\iota k_y)^2 \mathcal{F}_y \left[ p|_{i,:,l}^{n} \right] \right], \\
p_z|_{i,j,l}^{n+1} &= \frac{\sigma_{i,j,l}\delta_t - 2}{\sigma_{i,j,l}\delta_t + 2} p_z|_{i,j,l}^{n-1} + \frac{4}{\sigma_{i,j,l}\delta_t + 2} p_z|_{i,j,l}^{n} + \frac{2(c_{i,j,l}\delta_t)^2}{\sigma_{i,j,l}\delta_t + 2} \mathcal{F}_z^{-1} \left[ (\iota k_z)^2 \mathcal{F}_z \left[ p|_{i,j,:}^{n} \right] \right].
\end{aligned} \tag{7}
$$

We first update the acoustic projections by solving (7) and then compute the acoustic pressure at time $t^{n+1}$ by means of (6), that results in an explicit time-marching method. Where $\sigma_{i,j,l} = 0$, the scheme reduces to a classical

second-order in time explicit PSTD scheme for the second-order wave equation. In practical terms, DWE is applied to unbound wave propagation problems by assuming a zero $\sigma$ inside the physical domain slowly increasing its value as we approach the boundary. The larger the value of $\sigma$ the higher the absorption, although too steep a spatial change in $\sigma$ can lead to artificial reflections. Here, we follow Spa et al. (2014) considering a linear variation of $\sigma$ with respect to distance to the main grid of physical simulation, namely

$$\sigma_{i,j,l} = \sigma_0 \frac{d_{i,j,l}}{N_{\text{ABL}}}, \tag{8}$$

We remark that we have found the dimensionless quantity $\sigma_0 \delta_t$ better for the characterization of DWL absorption than $\sigma_0$, hence when calibrating DWL we will use $(N_{\text{ABL}}, \sigma_0 \delta_t)$ tuples to characterize different experiments for a fixed physical domain.

## 2.3 The sponge boundary layer (SBL)

Our second ABL under study is the SBL technique presented by Cerjan in Cerjan et al. (1985). The main formulation is based upon the second-order wave equation for pressure $p$, but also requires its temporal derivative $\dot{p}$ as an auxiliary dependent variable. The reason for this requirement is that part of the damping is applied directly





on $\dot{p}$. As a consequence, in PSTD we adopt a temporal staggered sampling of $p$ and $\dot{p}$, so that both variables are computed with central differences of $\delta_t$ step. The time marching algorithm consists on a two-step scheme, where

$\dot{p}$ is computed at the temporal half step $n+1/2$, for a subsequent computation of $p$ at the full step $n+1$. Similar to DWE, we also split both dependent variables into three Cartesian projections, and each projection is computed independently. The scheme starts with a first step

$$
\begin{aligned}
\dot{p}_x|_{i,j,l}^{n+\frac{1}{2}} &= \mu_{i,j,l} \cdot \dot{p}_x|_{i,j,l}^{n-\frac{1}{2}} + \mu_{i,j,l} \cdot c_{i,j,l}^2 \delta_t \cdot \mathcal{F}_x^{-1}\left[ (\iota k_x)^2 \, \mathcal{F}_x\left[p|_{:,j,l}^n\right]\right] \;, \\
\dot{p}_y|_{i,j,l}^{n+\frac{1}{2}} &= \mu_{i,j,l} \cdot \dot{p}_y|_{i,j,l}^{n-\frac{1}{2}} + \mu_{i,j,l} \cdot c_{i,j,l}^2 \delta_t \cdot \mathcal{F}_y^{-1}\left[ (\iota k_y)^2 \, \mathcal{F}_y\left[p|_{i,:,l}^n\right]\right] \;, \\
\dot{p}_z|_{i,j,l}^{n+\frac{1}{2}} &= \mu_{i,j,l} \cdot \dot{p}_z|_{i,j,l}^{n-\frac{1}{2}} + \mu_{i,j,l} \cdot c_{i,j,l}^2 \delta_t \cdot \mathcal{F}_z^{-1}\left[ (\iota k_z)^2 \, \mathcal{F}_z\left[p|_{i,j,:}^n\right]\right] \;,
\end{aligned}
\tag{9}
$$

where $\mu_{i,j,l}$ is a space-dependent absorption parameter, defined below, whereas the second step reads

$$
\begin{aligned}
p_x|_{i,j,l}^{n+1} &= \mu_{i,j,l}\left[ p_x|_{i,j,l}^n + \delta_t \cdot \dot{p}_x|_{i,j,l}^{n+\frac{1}{2}}\right] \;, \\
p_y|_{i,j,l}^{n+1} &= \mu_{i,j,l}\left[ p_y|_{i,j,l}^n + \delta_t \cdot \dot{p}_y|_{i,j,l}^{n+\frac{1}{2}}\right] \;, \\
p_z|_{i,j,l}^{n+1} &= \mu_{i,j,l}\left[ p_z|_{i,j,l}^n + \delta_t \cdot \dot{p}_z|_{i,j,l}^{n+\frac{1}{2}}\right] \;.
\end{aligned}
\tag{10}
$$

Equations (9) and (10), followed by the step given by Eq. (6), result in an explicit time-marching scheme. In the present work we follow Cerjan et al. (1985), to define values of $\mu_{i,j,l}$ as follows

$$
\mu_{i,j,l} = e^{-(\mu_0 \cdot d_{i,j,l})^2} \;,
\tag{11}
$$

where $\mu_0$ is SBL's dimensionless absorbing parameter. We will explore $(N_{\mathrm{ABL}}, \mu_0)$ tuples for a fixed physical domain in upcoming sections.

### 2.4  The Perfectly Matched Layers (PML)


The PML's formulation Bérenger (1994) requires first derivatives of the absorbed variables, in our case: pressure $p$ and velocity $\mathbf{v}$. The first-order Euler formulation of the wave equation (2) involves all directional spatial derivatives required by the PML implementation, thus it is natural to adopt this formulation for PML. The PSTD-PML method is a two-step time-staggered marching algorithm, that first updates the particle velocity components,

$$
\begin{aligned}
v_x|_{i+\frac{1}{2},j,l}^{n+\frac{1}{2}} &= \frac{1}{1+\alpha_{i,j,l}^x \delta_t}\left( v_x|_{i+\frac{1}{2},j,l}^{n-\frac{1}{2}} - \frac{\delta_t}{\rho} \cdot \mathcal{F}_x^{-1}\left[\iota k_x \mathcal{F}_x\left[p|_{:,j,l}^n\right]\right]\right) \;, \\
v_y|_{i,j+\frac{1}{2},l}^{n+\frac{1}{2}} &= \frac{1}{1+\alpha_{i,j,l}^y \delta_t}\left( v_y|_{i,j+\frac{1}{2},l}^{n-\frac{1}{2}} - \frac{\delta_t}{\rho} \cdot \mathcal{F}_y^{-1}\left[\iota k_y \mathcal{F}_y\left[p|_{i,:,l}^n\right]\right]\right) \;, \\
v_z|_{i,j,l+\frac{1}{2}}^{n+\frac{1}{2}} &= \frac{1}{1+\alpha_{i,j,l}^z \delta_t}\left( v_z|_{i,j,l+\frac{1}{2}}^{n-\frac{1}{2}} - \frac{\delta_t}{\rho} \cdot \mathcal{F}_z^{-1}\left[\iota k_z \mathcal{F}_z\left[p|_{i,j,:}^n\right]\right]\right) \;,
\end{aligned}
\tag{12}
$$





to finally compute the values of each projection of the acoustic pressure,

$$
\begin{aligned}
p_x|_{i,j,l}^{n+1} &= (1 - \alpha_{i,j,l}^x \delta_t) \cdot p_x|_{i,j,l}^n - \rho c_{i,j,l}^2 \delta_t \cdot \mathcal{F}_x^{-1}\left[\iota k_x \mathcal{F}_x\left[v_x|_{:,j,l}^{n+\frac{1}{2}}\right]\right], \\
p_y|_{i,j,l}^{n+1} &= (1 - \alpha_{i,j,l}^y \delta_t) \cdot p_y|_{i,j,l}^n - \rho c_{i,j,l}^2 \delta_t \cdot \mathcal{F}_y^{-1}\left[\iota k_y \mathcal{F}_y\left[v_y|_{i,:,l}^{n+\frac{1}{2}}\right]\right], \\
p_z|_{i,j,l}^{n+1} &= (1 - \alpha_{i,j,l}^z \delta_t) \cdot p_z|_{i,j,l}^n - \rho c_{i,j,l}^2 \delta_t \cdot \mathcal{F}_z^{-1}\left[\iota k_z \mathcal{F}_z\left[v_z|_{i,j,:}^{n+\frac{1}{2}}\right]\right].
\end{aligned}
\tag{13}
$$

Together with equation (6) we have a complete updating scheme. Above, the space-dependent parameter $\boldsymbol{\alpha} = (\alpha^x, \alpha^y, \alpha^z)$ is the vector quantity that controls PML absorption. Contrary to DWE or SBL, whose absorption depends, locally, only on the nodal distance to the main grid, in PML the outward direction from the physical domain is equally relevant. Similar to DWE (see eq. 8), we define a linear increase of $\boldsymbol{\alpha}$ components up to a maximum absorbing value $\alpha_0$, i.e.

$$
\boldsymbol{\alpha}_{i,j,l} = \alpha_0 \frac{d_{i,j,l}}{N_{ABL}} \hat{\mathbf{d}}_{i,j,l},
\tag{14}
$$

where $\hat{\mathbf{d}}$ is the unit vector from $(i,j,l)$ to the closest point in the physical domain's grid, namely

$$
\hat{\mathbf{d}}_{i,j,l} = \left(\frac{d^x}{d_{i,j,l}}, \frac{d^y}{d_{i,j,l}}, \frac{d^z}{d_{i,j,l}}\right).
\tag{15}
$$

Similar to DWL, we remark that we have found the dimensionless quantity $\alpha_0 \delta_t$ to be better at characterizing PML absorption than $\alpha_0$, hence when calibrating PML we will use $(N_{\text{ABL}}, \alpha_0 \delta_t)$ tuples to characterize different experiments for a fixed physical domain.

Finally, we would like to mention that the velocity-pressure scheme (12) and (13) is stated on a staggered spatial mesh, where shifting the spectral derivatives is critical to eliminate artifacts produced by the source generation, as previously reported in Ozdenvar and McMechan (1996).

## 2.5 Stability Bound and Dispersion Error

Before our application exercises, we briefly comment on the stability of PSTD and its dispersion properties. At uniform grids and using second-order explicit time integration, a Von-Neumann analysis of PSTD in unbounded acoustic media yields the following bound for conditional stability

$$
S = \max\{c_{i,j,l}\}\frac{\delta_t}{\delta} \leq \frac{2}{\pi\sqrt{3}},
\tag{16}
$$

In (16), $S$ is the Courant-Friedrichs-Lewy (CFL) number. In the case of a homogeneous medium $c_{i,j,l} = c$, this theoretical analysis also leads to the following expression for dispersion errors

$$
\frac{c_{\text{num}}}{c} = \frac{\pi\delta_t}{T\sin\left(\frac{\pi\delta_t}{T}\right)}.
\tag{17}
$$

Above, $c_{\text{num}}$ is the numerical wave speed and $T$ is the period of the given plane wave used in the Von-Neumann analysis. Thus, the spatial and temporal grid samplings must fulfill the inequality in (16) to guarantee stable





simulations. However, the numerical accuracy of PSTD simulations is mainly driven by the dispersion errors quantified by (17), which only depend on the time step. As a consequence, in practical PSTD applications, the spatial step can be fixed to the largest value allowed by the Nyquist sampling limit, but the time step must be taken much smaller than the one allowed by the stability bound, in order to control dispersion anomalies. In
other words, low-dispersive accurate PSTD simulations can be achieved using optimal $S$ values, which are far below the limit established by Eq. (16). These Von-Neumann analytical results and suitable choices on space and time resolution are reported in the broad literature on PSTD methods (e.g., Gazdag (1981); Fornberg (1998, 1987), and also Spa et al. (2020) for a recent review).

The coupling of the ABL techniques presented above to a PSTD method does not alter the stability and disper-
sion properties of PSTD in lossless unbound acoustic media. The physical attenuation experienced by acoustic waves at any frequency along the ABL regions only reinforces the boundedness of the numerical solution and thus favors the damping of short period oscillations induced by dispersion.

## 3    Calibration of ABL Parameters

In the previous Section we have written the formulations of all three ABL and remarked that two main param-
eters control absorption in each of them. Namely, the size of the absorbing layer $N_{\mathrm{ABL}}$, which is a parameter shared by all ABLs, and a specific parameter that depends on each ABL, namely $\sigma_0$, $\mu_0$ and $\alpha_0$ for DWE, SBL and PML, respectively. In the case of DWL and PML the absorption parameters have dimension of inverse time, thus in order to analyze absorption in a dimensionless framework we will use the tuples $(N_{\mathrm{ABL}}, \sigma_0 \delta_t)$, $(N_{\mathrm{ABL}}, \mu_0)$ and $(N_{\mathrm{ABL}}, \alpha_0 \delta_t)$ for DWE, SBL and PML, respectively. Our study aims at characterizing the absorption profiles,
namely absorption as a function of the tuples described above, of all three ABLs by means of experimentation. On homogeneous media, several authors have explored absorption parameter optimization through formal reflectivity and transmission analyses, for a particular ABL technique. For instance, Israeli and Orszag (1981); Kosloff and Kosloff (1986); Spa et al. (2014) formally study damping profiles in the case of DWE, while analyses on PML parameterization for elastodynamics can be found in Chew and Liu (1996); Collino and Tsogka (2001). For seis-
mic wave propagation, Gao et al. Gao et al. (2017) compare the empirical performance of different absorbing techniques on acoustic heterogeneous test cases using Finite Difference methods.

### 3.1    Methodology

Our characterization effort involves 1) finding appropriate tests for which a reference exists, 2) finding suitable metrics that measure the absorption performance of the methods against the reference, 3) establishing absorp-
tion thresholds that classify the absorption and 4) for each classification and ABL, finding the parameter tuple that requires less absorption nodes or $N_{\mathrm{ABL}}$. We will refer to such tuple, for each ABL, as the *optimal* tuple. In this sense optimality refers to reaching the desired absorption with the minimum possible number of grid points.



The first step to create an absorption measure is quantifying the total energy in the physical domain (not including the ABL) at any given time sample. Thus, we define the following quantity,

$$E|^n = \sum_{i=1}^{N_x} \sum_{j=1}^{N_y} \sum_{l=1}^{N_z} (p|_{i,j,l}^n)^2 \,,$$ (18)

and the corresponding dimensionless proxy,

$$\hat{E}|^n = \frac{E|^n}{\max\limits_{n \in [0,N_t]} E|^n} \,,$$ (19)

where the energy $E|^n$ is normalized by the maximum energy value present in the problem. Another key ingredient to create an absorption measure consists on building a proper reference signal, i.e. namely $\hat{E}_{REF}$. Let us assume that this signal can be constructed whether numerical simulations or analytical expressions. In any case, the following quantity is defined,

$$\Delta \hat{E} = \sum_{n=n_0}^{N_t} \left| \hat{E}_{REF}|^n - \hat{E}|^n \right| \delta_t \,.$$ (20)

Note that $n_0$ can be any value within the discrete time interval and its value, as well as the computation of $\hat{E}_{REF}$, would be obtained depending on the specifications of the problem. For example, in problems where it is impossible to characterize the energy via analytical expresions, we will use numerical simulations to compute the reference solutions in the whole discrete time range, i.e. $n_0 = 0$. In these cases, when the scenarios and the source characterizations are complex, we will build reference solutions by considering simulations with large number of ABL compared to the original simulation carried out to obtain $\hat{E}$. This way, we ensure lower contributions due to boundary reflexions getting an idea about the sensibility on the ABL implementation with respect to the number of absorbing nodes, $N_{ABL}$. In other words, $\Delta \hat{E}$ provides information on the differences between two signals, the computed signal, $\hat{E}$ and the reference signal, $\hat{E}_{REF}$. It means that low values of $\Delta \hat{E}$ represent strong similarities on both signals whereas high values of $\Delta \hat{E}$ exhibit differences between them.

On the other side, for problems where the domain has a constant propagation velocity, $c$, and the energy is injected by means of a source that is punctual and finite in time. If we know when the source stops injecting energy and when the energy inside the physical domain must be zero (the time iteration $n_0$), we can assume that $\hat{E}_{REF}|^n = 0$ for $n \geq n_0$ and, therefore, we define,

$$\epsilon = \log(\Delta \hat{E}) = \log \left( \sum_{n=n_0}^{N_t} \hat{E}|^n \delta_t \right) \,.$$ (21)

Instead of Eq. (20) that would be a measure of similarity between two signals, $\epsilon$ represents the remanent energy obtained due to the ABL approximation. In fact, it is worth pointing out that, under these conditions, the energy inside the physical domain at $n \geq n_0$ should be null and, cosenquently, it means that Eq. (21) provides direct





information about the absorbing features of the ABL implementation. Note that, in the next section, the calibration of the ABL approximations has been done by using the $\epsilon$ definition through Eq. (21). This way, we are able to measure the absorption performance of the three different methods under a same reference solution.

Finally mention that, for all scenarios in the following, the main grid remains identical and we modify only the size of the ABL zone and its associated absorption parameter. We will always exploit the spatial discretization characteristics of PSTD, thus using the coarsest grid possible at 2 points per minimum wavelength (ppw). All sources used will be point sources in space and Ricker wavelets in time. All numerical experiments that follow use our bespoke PSTD-ABL implementations using the g++ C compiler version 4.5.3.1-1, under -lm and -O3 optimization flags, and linking the FFTW3 library version 3.3.4-2. All simulations have been performed by an Intel Core i7-6820HK processor running at 2.70GHz under the Linux operating system.

### 3.2 Calibration for a homogeneous cube

We first consider a cube of size $500 \times 500 \times 500$ m³ with a constant wave velocity $c_{i,j,l} = 2000$ m·s⁻¹, and place a point source at the central location. The source time function is a Ricker wavelet with peak at $10$ Hz, and hence a maximum frequency of $\approx 25$ Hz, that excites a wavefield of minimum wavelength $\lambda_{\min} \approx 80$ m. We use a grid step of $\delta = 40$ m and a temporal step of $\delta_t = 0.002$ s, thus ensuring a stability number $S = 0.1$ that is less than $30\%$ of the stability limit. For this example, the wavefields leave the main grid at $n_0 = 208$. This specific value of $n_0$ results from the maximum travel time from source to the corners of the domain ($108$ time steps) and the time needed to finish injecting $95\%$ of the energy from the source wavelet ($100$ time steps). After $n_0$ the remaining energy in the domain comes from reflections at the ABL. As mentioned, we use Eq. (21) to measure the absorbing perfomance of the ABL implementations.

Next, we perform a numerical exploration of the $N_{\mathrm{ABL}}$-absorption parameter pairs, using the samples in Table 1. For each ABL, we vary both $N_{\mathrm{ABL}}$ and their respective absorption parameter, thus resulting in 620 scenarios per ABL.

**Table 1.** Sampling of the absorption parameters and absorbing layer size used for parameter exploration, for each ABL

|  | $N_{\mathrm{ABL}}$ | $\sigma_0 \delta_t$ (DWE) | $\mu_0$ (SBL) | $\alpha_0 \delta_t$ (PML) |
|---|---|---|---|---|
| min | 4 | 0.001 | 0.001 | 0.001 |
| max | 34 | 0.31 | 0.0414 | 0.61 |
| increment | 1 | 0.0163 | 0.0021 | 0.032 |
| samples | 31 | 20 | 20 | 20 |

Fig. 2 depicts $\epsilon$ values for the parameter ranges considered in Table 1 that include results for DWE (a), SBL (b) and PML (c) techniques. In the case of PML we restrict the vertical axis to $N_{\mathrm{ABL}} \leq 16$ as this results in already sufficient absorption of wavefields. For all cases there is an increase of absorption with $N_{\mathrm{ABL}}$ and we have a





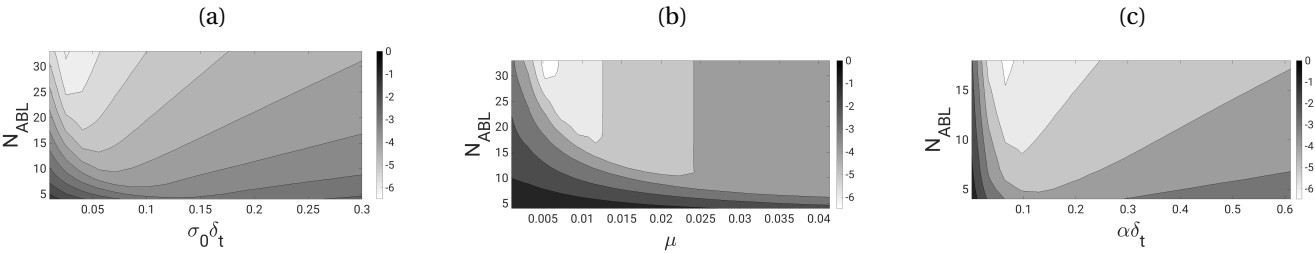

**Figure 2.** The grey scale depicts $\epsilon$ as a function of $N_{\mathrm{ABL}}$ and the absorbing parameters for (a) DWE, (b) SBL and (c) PML. Light greys indicate better absorption. Notice the smaller number of absorbing layers (vertical axis) used by PML.

window of optimal absorption parameters which depends mildly on $N_{\mathrm{ABL}}$. All ABL methods reach an absorption performance of $\epsilon < -6$ in the explored $N_{\mathrm{ABL}}$ range. PML is the most efficient technique because only requires $N_{\mathrm{ABL}} = 16$ to achieve this $\epsilon$ threshold. Alternatively, DWE reaches the same absorption using $N_{\mathrm{ABL}} = 32$, while

SBL employs $N_{\mathrm{ABL}} = 30$. Both PML and DWE absorption improve consistently with $N_{\mathrm{ABL}}$. Conversely, SBL seems less sensitive to increasing $N_{\mathrm{ABL}}$ and absorption seems to saturate after a given $N_{\mathrm{ABL}}$ value.

Table 2 shows, for several $\epsilon$ thresholds, which is the minimum $N_{\mathrm{ABL}}$ and coupled absorption parameter value. In the threshold range $-6 < \epsilon < -3.5$, DWE and PML deliver comparable accuracy, but the former needs at least twice the $N_{\mathrm{ABL}}$ value than the latter. Relative to DWE, SBL achieves same absorption for nearly similar $N_{\mathrm{ABL}}$.

Finally, Fig. 3 compares the time evolution of our energy proxy $\hat{E}|^n$ for each ABL technique, for three $\epsilon$ threshold values given in Table 2. For each ABL technique, significant differences on the $\hat{E}$ magnitude among these three curves are early observed, soon after $n_0$ iterations. In the DWE and SBL cases, large differences of the absorption efficiency persist during all $N_t = 1000$ iterations, but slighter differences are observed in PML curves. In next section, these three reference parameter sets will be exercised and compared, in ABL applications to heterogeneous

test cases.

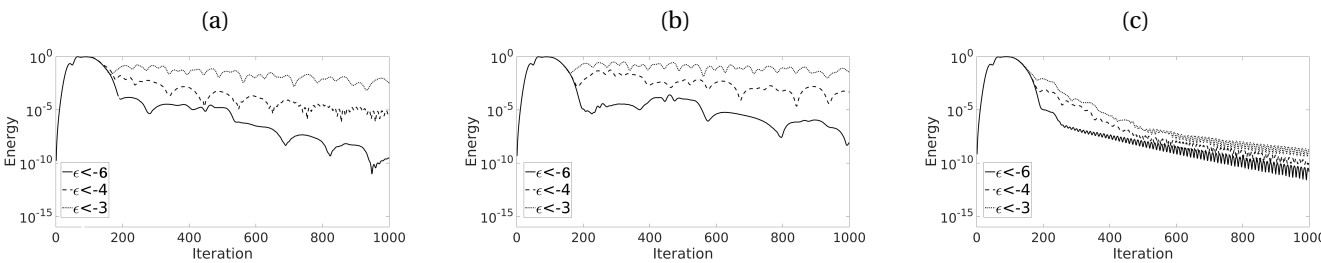

**Figure 3.** The time evolution of the energy proxy $\hat{E}|^n$ in logarithmic scale for the (a) DWE, (b) SBL and (c) PML techniques. ABL parameters are set as per Table 2.




**Table 2.** Optimal pairs of $N_{\mathrm{ABL}}$ and associated ABL parameter found for each $\epsilon$ threshold.

| | DWE | | SBL | | PML | |
|---|---|---|---|---|---|---|
| Accuracy | $N_{\mathrm{ABL}}$ | $\sigma_0 \delta_t$ | $N_{\mathrm{ABL}}$ | $\mu_0$ | $N_{\mathrm{ABL}}$ | $\alpha_0 \delta_t$ |
| $\epsilon < -3$ | 5 | 0.086 | 7 | 0.031 | 4 | 0.065 |
| $\epsilon < -3.5$ | 7 | 0.071 | 8 | 0.030 | 4 | 0.097 |
| $\epsilon < -4$ | 10 | 0.056 | 11 | 0.020 | 5 | 0.097 |
| $\epsilon < -4.5$ | 14 | 0.041 | 14 | 0.016 | 6 | 0.097 |
| $\epsilon < -5$ | 18 | 0.041 | 17 | 0.012 | 9 | 0.097 |
| $\epsilon < -5.5$ | 25 | 0.025 | 23 | 0.007 | 12 | 0.065 |
| $\epsilon < -6$ | 32 | 0.025 | 30 | 0.005 | 16 | 0.065 |

### 3.3 Accuracy analysis for geophysical imaging

In addition to sheer energy absorption it is important to analyze the impact of our different ABL in practical imaging applications. As a simple yet representative test, we analyze a reverse time migration (RTM) case in a homogeneous model with a single source and receiver. In reverse-time migration (see e.g., Claerbout et al. (1985))

an image, or reflectivity map, of the subsurface is obtained by means of two seismic simulations. A forward simulation propagates the source wavelet signal through the domain of interest, whereas a backward simulation propagates the data recorded in the field for that same source, reverted in time. By correlating the wavefields of forward and (time-reversed) backward simulations we generate the image of the subsurface, which indicates regions of impedance in the subsurface that may have generated the observed data. RTM has the advantage over

other imaging modalities of supporting completely heterogeneous 3D velocity models, as well as incorporating all finite-frequency phenomena associated with acoustic waves, such as multiple reflections or scattering. On the other hand it is costly in terms of computation (it relies on simulations) and inherits all inaccuracies of the wavefield simulation algorithms (e.g. imperfect boundaries) which may result in artifacts in the image. A calibration exercise frequent to imaging, and specifically to RTM, is known as impulse response image Claerbout et al. (1985);

Ng (2007). In an impulse response a single-source image is generated by placing a single hypothetical receiver at the same location as the source. The receiver may include several well known pulses, which when *imaged* into the domain of interest, result in patterns that can be analyzed to assess how accurate images can be obtained at different, e.g. depths or frequencies. For a homogeneous-model impulse response, there is no preferred origin for the reflections, which are imaged as concentric half-spheres of finite width centered at the source/receiver locus.

Furthermore, the amplitude of the resulting image is independent on energy spread, and thus any disagreement between the expected and modelled image is due to modelling errors. In our case, we choose to investigate the





vertical image column of the impulse-response image that contains the source (and receiver). For such simple configuration it is easy to obtain an exact solution to the problem, and hence we can use a time-frequency analysis Kristekova et al. (2006) to check the quality of our image. Time-frequency analysis typically refers to temporal

signals. As in our case the image exists in the spatial domain, we can refer to an analogous space-wavenumber analysis.

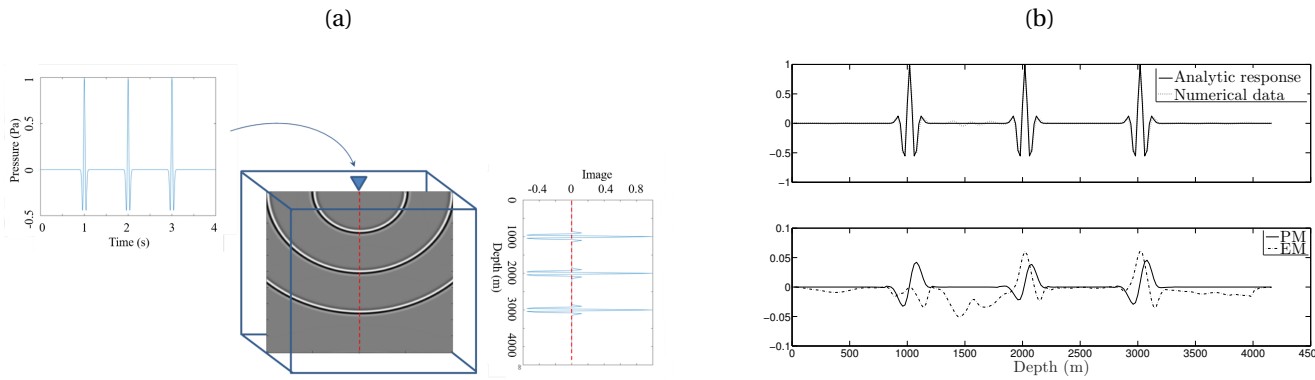

**Figure 4.** (a) Reflectivity image (in greyscale) from an impulse response in homogeneous media, with the receiver data in horizontal and a close-up of the exact image column in vertical. (b) Example of image column results compared to the exact reference (top) and the envelope and phase errors between them (bottom). The discrete sum of the error curves results in EM and PM respectively.

We use the same model, grid steps $\delta$ and $\delta_t$, and wavelet as in Section 3.2 with the following exceptions: the domain size is enlarged to $4 \times 4 \times 4 \, \text{km}^3$, and the source is placed at $(2, 2, 0)$ km. A receiver is located at the same point as the source. The data signal, in this impulse response study, contains three pulses, equal in shape to the

source wavelet, but with peaks at 1, 2 and 3 seconds, respectively (see Fig. 4 (a)). Given the homogeneous velocity 2000 m·s$^{-1}$, the data is mapped in the image as three concentric half-spheres centered at the source/receivers location and with radii 1, 2 and 3 km, respectively, see Fig. 4. These particular radii are the distances compatible with acoustic reflectors generating the data (i.e. three wiggles, at 1, 2 and 3 s). In order to assess the accuracy of the ABLs, we use the same ABL parameterizations obtained in Section 3.2 (see Table 2) and we measure both

envelope (EM) and phase (PM) misfits with respect to the reference solution.

Table 3 presents the results in terms of EM and PM with respect to the absorption configurations found in Table 2. As expected, and further validating the findings of Section 3.2, for high $\epsilon$ thresholds the misfits EM and PM are small. Both errors decrease monotonically for PML, whereas misfits delivered by SBL and DWE show some oscillations in the range $-6 < \epsilon < -5$. In all cases, we find SBL performing slightly better than both DWE

and PML, and in the case of highest absorption $\epsilon < -6$, its performance is comparable to PML for both PM and





**Table 3.** Envelope Misfits (EM) and Phase Misfits (PM) obtained when using the three ABL techniques under different $\epsilon$ accuracy thresholds. Both EM and PM are dimensionless quantities.

| DWE | EM | PM | SBL | EM | PM | PML | EM | PM |
|---|---|---|---|---|---|---|---|---|
| $\epsilon < -3$ | 0.2073 | 0.0930 | $\epsilon < -3$ | 0.1622 | 0.0701 | $\epsilon < -3$ | 0.1995 | 0.0727 |
| $\epsilon < -3.5$ | 0.1599 | 0.0919 | $\epsilon < -3.5$ | 0.1310 | 0.0498 | $\epsilon < -3.5$ | 0.1443 | 0.0563 |
| $\epsilon < -4$ | 0.1233 | 0.0705 | $\epsilon < -4$ | 0.1180 | 0.0460 | $\epsilon < -4$ | 0.1236 | 0.0494 |
| $\epsilon < -4.5$ | 0.1190 | 0.0674 | $\epsilon < -4.5$ | 0.1106 | 0.0444 | $\epsilon < -4.5$ | 0.123 | 0.0478 |
| $\epsilon < -5$ | 0.1184 | 0.0653 | $\epsilon < -5$ | 0.1146 | 0.0448 | $\epsilon < -5$ | 0.1175 | 0.0464 |
| $\epsilon < -5.5$ | 0.1309 | 0.0672 | $\epsilon < -5.5$ | 0.1147 | 0.0443 | $\epsilon < -5.5$ | 0.1149 | 0.0454 |
| $\epsilon < -6$ | 0.1234 | 0.0669 | $\epsilon < -6$ | 0.1134 | 0.0448 | $\epsilon < -6$ | 0.1146 | 0.0452 |

EM metrics. We can thus conclude that the parameterization pairs obtained in the previous sections result in better image accuracy as the absorption of the ABLs increases, i.e., as the $\epsilon$ threshold decreases.

As an additional comparison, we compute the same impulse response exercise using an algorithm popular in geophysical imaging: finite-differences with $8^{th}$ order in space, $2^{nd}$ order in time and using $\delta = 20$ m and $\delta_t = 0.003454$ s. In this case we obtain EM$\sim 0.14$ and PM$\sim 0.07$ when using $2^{nd}$ order Higdon paraxial ABCs. Both numbers can be matched, and improved, by using the algorithms presented here. Notice that the spatial grid of this alternative scheme is considerably larger that the one used with the PSTD method presented in this study, due to the higher points per wavelength needed in finite-difference schemes.

## 4 Validation of ABL parameters in heterogeneous media

The Earth's subsurface is largely heterogenous across many scales. In such environments wavefields become more complex, involving scattering, reflections and refractions, among other phenomena. As a consequence, a generalized calibration of ABLs is not possible, as all models are fundamentally different from each other. Our goal when studying ABLs in heterogeneous media is assessing whether their fundamental behaviour remains, i.e. absorption increases steadily with $N_{\text{ABL}}$, and if our calibration results, which were obtained for homogenous models, are also useful for heterogenous models. We remark that we will use a direct continuation strategy to populate velocity values at the absorbing layers, as defined in the first subsection of Section 2.

### 4.1 Three-layered Medium

First, we consider a 3D cuboid physical domain involving three flat layers of wave speeds 2000, 4000 and 6000 m·s$^{-1}$, respectively. The central layer is 1320 m thick, being the top and bottom layers both half-spaces. The





source parameters and the size of the domain is the same as in Section 3.2. However, in this test case, the Ricker point source is located $1300$ m above the first material interface, and therefore inside the top layer, but still central in the other two directions. We run simulations for $N_t = 5000$ iterations, for a total simulation time of $10$ s.

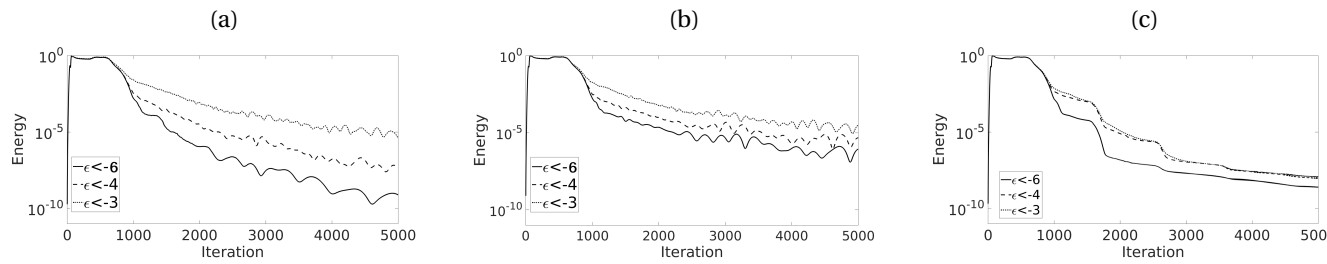

**Figure 5.** The time evolution of the energy proxy $\hat{E}|^n$ in logarithmic scale for the three-layered test using (a) DWE, (b) SBL and (c) PML. ABL parameters are set as per Table 2.

In Fig. 5, we show the evolution of our energy proxy $\hat{E}|^n$ during the simulation time. We observe $\hat{E}|^n$ diminishing for all cases after the first approximately 1000 iterations. The rate at which $\hat{E}|^n$ is reduced afterwards depends
on the ABL and the threshold used. We remark that the ABLs are parameterized following Table 2. Consistent with previous observations in Fig. 3 for homogenous media, lower $\epsilon$ thresholds result in better absorption. In addition, if we focus on long-term absorption (i.e. at iteration 5000 or $\hat{E}|^{5000}$), DWE at $\epsilon < -6$ reaches the smallest energy proxy values among all methods and configurations, whereas PML delivers small energy proxy values regardless of the parameter configuration chosen. DWE appears to be the most sensitive ABL to parameter changes, having
the largest difference between best and worse absorption among all methods tested.

### 4.2 The SEG-EAGE Salt Model

As a final and more realistic scenario, we use the 3-D SEG/EAGE Salt 3D model (see, e.g.Yoon et al. (2003)) and test our ABL for a modelling exercise. This model has been extensively used for benchmarking exercises in geophysics because it includes features typically observed in the subsurface. The model dimensions are $(7.5, 7.5, 3.6)$ km, and
we locate a point source at $(x_s, y_s, z_s) = (3.75, 3.75, 0)$ km. In this model, the wave speed varies from $1500$ m·s$^{-1}$ at the top water layer, to $4200$ m·s$^{-1}$ inside the salt body (see Figure 6). We add an ABL to each boundary of the physical model resulting in an unbounded domain. As in previous experiments, we use a Ricker source wavelet with a maximum frequency of $25$ Hz and adapt the grid spacing to $\delta = 30$ m to accomodate the model's minimum velocity. Similarly, the time discretization is $\delta_t = 0.002$ s, which results in a maximum stability number $S = 0.28$.
In this test case, PSTD simulations last for $4$ s, i.e., they involve $N_t = 2000$ time iterations. In order to quantify the absorption for such a complex model we need run several configuration of ABLs and compare to a reference. To construct such reference solution, we use the PSTD simulation that employs PML using the parameters associ-

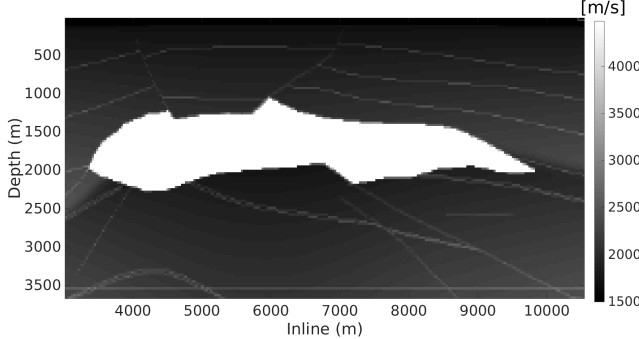

**Figure 6.** A vertical cross section, along the z-x plane located at y = 6800 m, of the 3D SEG/EAGE Salt velocity model. The white (high-velocity) part is a salt body.

ated with maximum absorption in Table 2 and $N_{\mathrm{ABL}} = 120$. Absorption for each ABL is next quantified using this reference. As illustration, Fig. 7 compares the wavefield at time step 430 ($t$=0.86s) obtained when using the best

DWE configuration reported in Table 2 with the reference solution. Specifically, snapshot at the left side uses DWE with $\sigma_0\delta_t = 0.0025$ and $N_{ABL} = 32$, whereas snapshot at the right side uses the reference PML configuration. The central figure shows the difference between these two snapshots at this time step. Most differences appear from a reflected wavefront by the top ABL in simulations using DWE. As the PSTD algorithm and simulation parameters are identical, i.e., time and grid stepping, these differences arise from the less effective absorption achieved by

DWE.

We run simulations using all 3 ABLs using all absorption parameter pairs reported in Table 2, and compute the corresponding errors using the metric defined in (20) for $n_0 = 0$ and the reference solution based on PML. The results are reported in Table 4 for all cases. For each ABL, errors steadily diminish with lower $\epsilon$ thresholds, i.e., as we sequentially employ the optimal parameters pairs given in Table 2. This is a remarkable result, as it confirms

the results from Section 3.2, i.e., we can use the calibration parameters obtained from a homogeneous case and observe improvements in absorption in a complex heterogeneous case. Results in Table 4 are also consistent with the absorption improvements achieved by using the three parameter choices employed in the previous three-layered medium test. Finally, please notice that under the same $\epsilon$ threshold, most PML errors in Table 4 are smaller than those reported by SBL, while DWE delivers the larger errors. However, the slightly lower efficiency

of DWE compared to SBL might be related to this particular SEG-EAGE model, and results can be reversed in a different seismic scenario, as already observed in the three-layered test (see, Fig. 5).

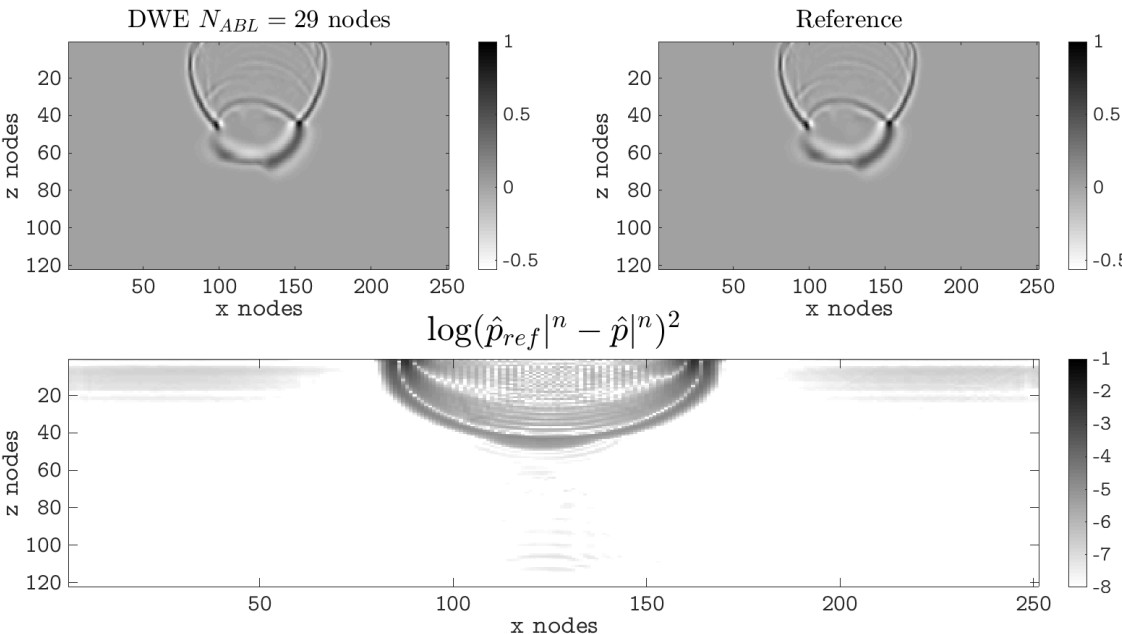

**Figure 7.** Snapshots of pressure at $t = 0.86$ seconds using DWE with $\sigma_0 \delta_t = 0.025$ and $N_{ABL} = 32$ at the top left and PML with $\alpha_0 \delta_t = 0.065$ and $N_{ABL} = 120$ at the top right. The bottom image shows their difference $(\hat{p}_{ref}|^n - \hat{p}|^n)^2$ in logarithmic scale.

**Table 4.** Errors $\Delta \hat{E}$ computed on the SEG/EAGE 3D Salt model using the absorption parameter pairs reported in Table 2.

| $N_{\mathrm{ABL}}$ | $\sigma_0 \delta_t$ | DWE | $N_{\mathrm{ABL}}$ | $\mu_0$ | SBL | $N_{\mathrm{ABL}}$ | $\alpha_0 \delta_t$ | PML |
|---|---|---|---|---|---|---|---|---|
| 5 | 0.086 | 0.0515 | 7 | 0.031 | 0.0419 | 4 | 0.130 | 0.1083 |
| 7 | 0.071 | 0.0291 | 8 | 0.030 | 0.0228 | 4 | 0.097 | 0.0277 |
| 10 | 0.056 | 0.0142 | 11 | 0.020 | 0.0106 | 5 | 0.097 | 0.0077 |
| 14 | 0.041 | 0.0083 | 14 | 0.016 | 0.0060 | 6 | 0.097 | 0.0028 |
| 18 | 0.041 | 0.0037 | 17 | 0.012 | 0.0036 | 9 | 0.097 | 0.0023 |
| 25 | 0.025 | 0.0033 | 23 | 0.007 | 0.0019 | 12 | 0.065 | 0.0013 |
| 32 | 0.025 | 0.0013 | 30 | 0.005 | 0.0010 | 16 | 0.065 | 0.0010 |

## 5 Comments on the Computational Times of ABL techniques

In this section, we discuss on the computational times obtained for our different ABLs coupled with PSTD acoustic wave simulations. Of course, observations in terms of compute time are less objective measures, because





times are affected by the algorithm design, compilation optimization, coding skills and libraries employed, hence we do not suggest that our findings are universal. Nevertheless, we will start our analysis with two theoretical aspects or considerations.

First we consider the memory footprint of PSTD using the three ABLs. As formulated in Section 2, our three ABLs require storage of 7 3D arrays. Each array covers the computational domain of size $(N_x + 2N_{\text{ABL}})(N_y +$

$2N_{\text{ABL}})(N_z + 2N_{\text{ABL}})$. In particular, DWE uses $p_x|^{n+1}, p_y|^{n+1}, p_z|^{n+1}, p_x|^n, p_y|^n, p_z|^n, p|^n$, SBL uses $p_x|^{n+1}, p_y|^{n+1},$ $p_z|^{n+1}, \dot{p}_x|^{n+1}, \dot{p}_y|^{n+1}, \dot{p}_z|^{n+1}, p|^n$ and PML uses $p_x|^{n+1}, p_y|^{n+1}, p_z|^{n+1}, v_x|^{n+1/2}, v_y|^{n+1/2}, v_z|^{n+1/2}, p|^n$.

Lastly we consider the amount of operations required per time update. Both DWE and SBL compute a single 1D spectral derivative of $p|^n$ along each coordinate, while PML computes an additional differentiation for each velocity component. Therefore, DWE and SBL benefit from the second-order linear wave equation formulation

and require half the number of Fourier transforms than the PML-based algorithm, which relies upon the first-order Euler formulation. Although the previous theoretical discussion considers the same number of absorbing

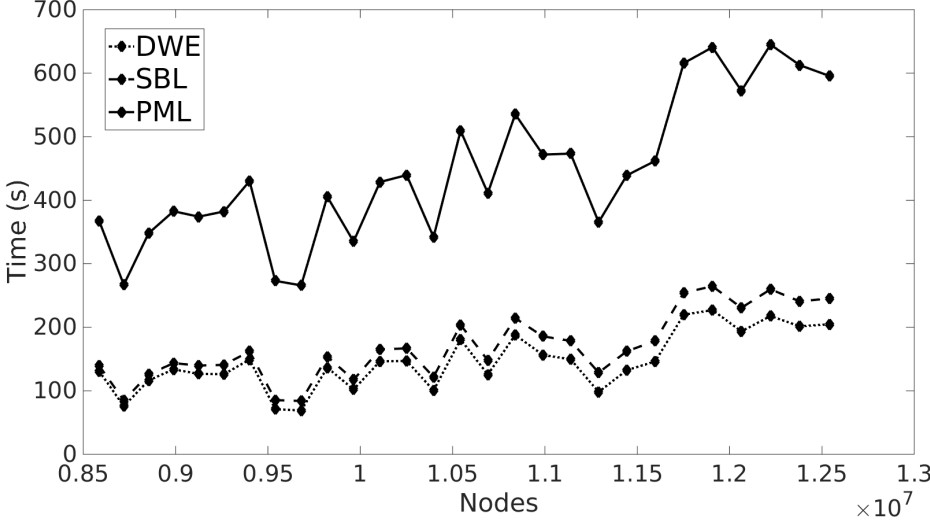

**Figure 8.** Computational time of all ABLs at different grids, characterized by their total number of nodes.

layers for all methods, we must recall that in Sections 3.2 and 4 we have consistently observed that PML requires about half the absorbing layers $N_{\text{ABL}}$ than either DWE or SBL for the same absorption. Nevertheless, given the usual size of geophysical domains, which are much larger than the number of layers considered in ABLs (i.e.

$N_x, N_y, N_z >> N_{\text{ABL}}$) this aspect does not result in a substantial advantage for PML in terms of memory or computational time.

Following the theoretical discussion, we have measured computational times for our PSTD code using the three ABLs for different grid sizes. In Figure 8, we present the computational times of 28 different grids using the

**Table 5.** Relative computing time $\tau$ with respect to the reference solution for the experiment 1 in Section 4.2. All ABL parameters follow $\epsilon$ thresholds in Table 2. Average, minimum and maximum times are included in the last three columns, respectively.

| Method | $\epsilon < -3$ | $\epsilon < -3.5$ | $\epsilon < -4$ | $\epsilon < -4.5$ | $\epsilon < -5$ | $\epsilon < -5.5$ | $\epsilon < -6$ | $\hat{\tau}$ | $\tau_{min}$ | $\tau_{max}$ |
|---|---|---|---|---|---|---|---|---|---|---|
| DWE | 0.0287 | 0.0549 | 0.0214 | 0.0627 | 0.0652 | 0.0961 | 0.0542 | 0.0547 | 0.0214 | 0.0961 |
| SBL | 0.0611 | 0.0557 | 0.0234 | 0.0399 | 0.0541 | 0.0821 | 0.0451 | 0.0516 | 0.0234 | 0.0821 |
| PML | 0.1329 | 0.1327 | 0.0778 | 0.1162 | 0.1285 | 0.0613 | 0.0906 | 0.1057 | 0.0613 | 0.1329 |

setup of the experiment 1 in Section 4. The total number of nodes in the grid is defined as $(N_x + 2N_{ABL})(N_y +$

$2N_{ABL})(N_z + 2N_{ABL})$ and $N_{ABL}$ ranges from $4$ to $31$. Three different conclusions can be drawn from the figure: 1) computational cost increases, on average, with grid size, as expected; 2) PML is approximately twice slower than either SBL or DWE for the same grid size and 3) there is an important variability in compute cost from the average trend, of about 15-20% with respect to the average value. The variability is very similar for all three ABLs at a single grid, hence stems from the node count and not other specific aspects of the different ABLs. This last result can

be surprising when compared to other computational methods such as finite-differences or finite-elements, but stems from the complex heuristics of modern FFT and DFT implementations, as will be further discussed later in this section. We remark that we use the FFTW3 library version 3.3.4-2 in our study.

To further expand our cost analysis we present results for our experiments in Section 4 comparing computational times for several ABL configurations relative to those obtained at the reference domain. Such relative time

metric is referred to as $\tau$. In Table 5 we show relative times for all parametric cases, as well as their average $\hat{\tau}$ and both minimum and maximum, i.e. $\tau_{min}$ and $\tau_{max}$, respectively, among all parameterizations used. In the Table, we observe that average times $\hat{\tau}$ for PML are about double than $\hat{\tau}$ for the other two ABL approaches, as expected from our previous analysis and consistent with Figure 8. For increasing absorption ranges in Table 5, we require $N_{ABL}$ to be larger for all ABL. Although in other seismic modelling methods this would result in a consistent

increase in compute time, this is not the case for PSTD. Compute times are rather spread and do not increase monotonically with respect to $\epsilon$ thresholds. The explanation for this result, consistently with what is observed in Figure 8, is the following: Novel FFT libraries rely on different factorizations and algorithms in order to optimize time to solution, for each node count. This results in FFTs that are very fast but also highly susceptible to significant variations as a function of the sample/node count. We rely on FFTW3 in our case, but similar behaviour

is observed in other contemporary FFT libraries (see, e.g. Khokhriakov et al. (2018) for an example using Intel MKL) and should be considered normal for PSTD or other Fourier-based methods. As a final recommendation, given the small value of $N_{ABL}$ with respect to the main grid dimensions in geophysical applications, it might be beneficial to test different $N_{ABL}$ values to reduce computational cost while keeping similar absorption.



## 6 Conclusions

In this work, we have reviewed and compared the three main ABL methodologies available in the context of PSTD simulations for acoustic wave propagation. Specifically, the damped wave equation (DWE), the sponge boundary layer (SBL) proposed in Cerjan et al. (1985), and a classical split perfectly matched layer (PML) formulation, have been developed and their algorithms outlined. The three ABL are relevant because they allow us to keep a pure Fourier pseudospectral scheme, without hybrid approximations at the boundaries. Absorption of DWE, SBL and

PML is controlled by the number of layers $N_{\mathrm{ABL}}$ and a single parameter specific to each formulation, i.e., $\sigma_0 \delta_t$, $\mu_0$ and $\alpha_0 \delta_t$ for DWE, SBL and PML, respectively. We have performed a calibration study on a simple homogeneous medium, extracting optimal configurations (i.e., those with minimum boundary size $N_{ABL}$) for a series of energy absorption thresholds. To that goal, Such configurations have been put to the test in a series of exercises of different heterogeneity distributions and complexity. We have established that configurations that resulted in high

absorption in our calibration, which involved a cube with homogeneous properties and just measured reflected energy, allow us to: 1) obtain better quality in a seismic imaging exercise, both in terms of phase and amplitude 2) achieve better absorption also in a three-layered model, despite the change in space/time sampling required by the heterogeneity and the more complex wavefields involved such as reflections and refractions and 3) accomplish better absorption in a complex 3D heterogeneous case. Hence, we can conclude that the configurations

obtained in our simple calibration study lead to increased quality of results for all cases tested. Such configurations are meant to be guidelines for modelling or imaging practitioners which can then be specialized to fit their accuracy needs.

   Comparing the three ABLs with each other is a complex issue. On one hand, DWE and SBL have very similar formulations and behave similarly in terms of $N_{ABL}$ for a given absorption threshold and computational cost.

On the other hand, PML requires fewer boundary layers for the same absorption level at the price of a higher overall computational cost, approximately double than DWE and SBL. Among these ABL methods, SBL presents less sensitivity to the increment of $N_{ABL}$.

   To assess absorption performance, we have introduced a dimensionless measure proportional to the total acoustic energy in the seismic volume, and use its magnitude in the calibration of ABL parameters. This en-

ergy proxy is consistent with the reflected energy that we qualitatively observe in all test scenarios, and therefore, we recommend it for similar studies of absorbing methods.

   We remark that compute times increase with grid size, but not in a steady or monotonic behaviour, as a result of using modern FFT libraries. Therefore varying the absorption of ABLs by means of larger $N_{ABL}$ values does, unintuitively, not necessarily result in increased computational time. Therefore, compute times are not strictly

predictable other than PML being significantly more expensive in terms of compute time than either DWE or SBL.



*Code and data availability.* Computer codes to run all three test cases are readily available at the Zenodo site https://doi.org/ 10.5281/zenodo.8113480 along with a README file to guide code compilation and execution. The input dataset for the EAGE SEG-SALT test case is available at the Zenodo site https://doi.org/10.5281/zenodo.7821703.

*Author contributions.* CS and JP implemented computer codes and carried out simulations. CS, JP and OR developed the mathematical formulation and designed test cases. CS and OR worked on the document editing.

*Competing interests.* The authors declare that they have no conflict of interest.

*Acknowledgements.* This project has received funding from the European Union's Horizon 2020 research and innovation pro-gramme under the Marie Sklodowska-Curie grant agreement No 777778 MATHROCKS. In addition, the research leading to
these results has received funding from the QUSTom project with proposal number 101046475 under the call HORIZON-EIC-2021-PATHFINDEROPEN-01.





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
