# Peer review of "Calibration of Absorbing Boundary Layers for Geoacoustic Wave Modeling in Pseudo-Spectral Time-Domain Methods"

_Geoscientific Model Development, 2023_

## Author Comment (AC1)

Dear Reviewer,

In this letter we answer the comments and we correct the small points also suggested.

First, we focus on the comments:

**Q1: If I understand it correctly, the error measure considers the wavefield in the entire domain. In many applications, however, one would only be interested in a few measurement points at the surface. Have you done any analysis using such a restricted error criterion, and would you expect a qualitative change of the results?**

A1: Yes, you are correct. We haven't focused on a specific area, because in many applications (e.g. migration, inversion) quality is required for the complete domain, and not just at the surface. We do not expect significant differences between the subset of nodes at the surface and the rest, for long simulations.

**Q2: You mention in section 2.1 that by setting the ABL parameter to zero, the same wave equation can be used in the physical domain and in the exterior. Do you distinguish those domains in the implementation? This would potentially affect computational performance considerably (with potential implications on load balancing)?**

A2: We use a monolithic approach to solve the same equation for each node, only that in the interior nodes a zero multiplies the absorbing terms. It definitely affects performance, but we didn't want to go deeper into this subject.

We clarify this point including this text around line 440 of the new document:

"Finally, we remark that for all methods, we solve the complete absorbing equation for each grid node, only using non-zero values for the absorbing parameters inside the absorbing layers."

**Q3: I understand that absorbing boundaries for elastic media are a totally different can of worms, but I would be interested in getting your take on the validity / transferability of the results to VTI acoustic (and maybe elastic) media.**

A3: Yes, this is part of our future work. We definitely need to re-run the calibration process for each physical setup to test that 1) the methodology holds and 2) which specific set of parameters are found to be optimal.

To address this point, we write this text at the end of the third paragraph in the conclusions section:

"The methodology to calibrate ABLs in this work could be applied to other wave equations such as the elastic wave equation or anisotropic wave equation. We do not expect the same calibration values to hold across all the equations, but the methodology should reveal the optimal values for each case. This will be subject of future work"

**There just a few more small points I noted, which I am listing below:**

**Line 100: Do you assume differentiability of the source in time or in time and space? Just wondering about point sources here.**

We use a regularization of Dirac's delta function for the spatial component of point sources, which is a gaussian. In time we chose a Ricker wavelet which is the second derivative of a gaussian. We atttempt, with this process to avoid contributions beyond the Shannon-Nyquist sampling theorem.

To clarify this point, we write at the end of the first paragraph of Section 3.2 this text:

"Note that we use a regularization of Dirac's delta function for the spatial component of point sources, which is a gaussian. In time we chose a Ricker wavelet which is the second derivative of a gaussian. We attempt, with this process to avoid contributions beyond the Shannon-Nyquist sampling theorem."

**Line 125: What is the reasoning behind adding the additional point with a pressure of zero? Is it correct that this essentially gives a homogeneous Dirichlet boundary condition? Why don't you use some first-order condition at the outer boundary instead?**

In Fourier spectral methods, we need to ensure continuous periodicity of the spatial distributions. By imposing exact zero values this condition is met. This is not sufficient to avoid artifacts but necessary to mitigate the Gibbs phenomenon. First-order conditions would not be able to ensure periodicity in the same way.

At the end of the first paragraph of Section 2.1,we write this text:

"It is important to remark that these extra nodes are essential to avoid the Gibbs phenomenon at the edges of the spatial mesh. Note that spectral derivatives require imposing periodicity to  the spatial distributions, therefore in this way, we ensure spatial periodicity in any direction of the mesh."

**Line 160: Have you analyzed the effect of the non-differentiability of sigma, i.e., the kink at the transition from the inner to the outer domain? I can imagine this might lead to artificial reflections.**

No, we have used only a linear profile which indeed results in impedance. However, all discrete versions of these ABLs incur in impedance at the boundary or inside the layer. For a detailed analysis of higher-order profiles we recommend Spa et al (2014).

To clarify this point, we add this text at the end of the final paragraph of Section 2.4:

"Finally, it is worth to mention that there exist other profiles that perform better, see for instance \cite{Spa2014} that they suggest order $3$ and $4$ polynomial absorbing profiles. However, in this analysis, we chose a linear profile because we prefer to focus on both, the calibration methodology and the design of the numerical experiments, rather on studying  specific absorbing profiles of each method."

**Eq. 11: Is there any physical intuition behind the exponential decay? I assume artificial reflections can occur similar to DWE if the slope is too steep? Intuitively, I would have assumed that N_ABL also enters into the formula, but it seems the values at the outer boundary will differ depending on the distance only? Just as a personal preference, I probably would have used other symbols than $\sigma$ and $\mu$ for the ABL-related coefficients, as I would associate those with stress and shear modulus, respectively.**

We are just following the standard reference here (Cerjan 1985). As we mentioned in a previous answer, we do not focus our attention on particular profiles, but rather on a methodology to calibrate the main parameters. Definitely, there should be a dependence between the parameter and

N_ABL. However, as our methodology always analyzes pairs of N_ABL and the parameter, such dependence loses relevance, at least for our purposes.

At the end of the section 2.3, we write this paragraph to clarify this point:

"It is important to mention that this profile is neither polynomial nor dependent on $N_{\mathrm{ABL}}$. As we mentioned in the previous subsection, we do not focus our attention on particular profiles, but rather on a methodology to calibrate the main parameters. Definitely there should be a dependence between the parameter and $N_{\mathrm{ABL}}$. However, as our methodology always analyzes tuples of $N_{\mathrm{ABL}}$ and the parameter, such dependence loses relevance, at least for our purposes."

**Line 265: Typo: expressions**

Corrected.

**Line 288: Out of curiosity: Is there a reason for not using a more recent version of g++?**

No, we just used the version that we had installed at that moment. We believe that only relative performance between runs should be used, being the absolute performance dependent on the hardware and the compiler, as well as potential optimizations carried out by the authors. Hence we only report relative times.

**Line 397: Just to double-check: Are you using a free surface condition at the top? I don't think this is the case, but I somehow would have expected this for the realistic SEG/EAGE model.**

No, we are not using a free surface. We have focused on the infinite case. We believe that adding a free-surface should require a complete re-calibration of the parameters. We leave this for a future work.

We add this text into the first paragraph of section 4.2:

"We remark that we are not adding a free surface condition to be compatible with the calibration exercise of the previous sections which also were unbounded."

**Line 445: Consistency when referring to Fig., Figure, figure.**

Checked

**Line 480: Typo: Such -> such.**

Checked

---

## Author Response (AR1)

Dear Editor:

According to your major and minor comments, we have done the following answers:

**Major Comments**

**Q1.-Can the authors discuss the applicability of this ABL analysis to wider classes of wave equations. For example, dispersive and/or nonlinear wave equations? Besides geoacoustics, fields such as biomedical acoustics and electromagnetics, often use dispersive and nonlinear models.**

A1.- It would be adventurous to extrapolate directly the analysis in the present work to other methods without conducting actual experiments to support them. However, as requested, we can discuss on the subject. On one hand, the method itself does not make assumptions regarding the underlying PDE or numerical solver employed. On the other hand, it is fairly simple and general, involving calibration 1) on representative models, 2) using well-defined metrics, 3) involving just two parameters. Therefore we believe that the method has potential for broad application without significant modifications, which will be the subject of future research by the authors.

**Q2.- While the PSTD is widely used in wave propagation problems, other numerical methods are also available, such as spectral finite element methods and finite volume methods (amongst others). Are any of these results applicable to such methods. For example, is the efficiency of the SBL relative to the PML restricted to PSTD, or is it a more general result? While I don't expect the authors to implement ABLs in such solvers, a short discussion of the wider applicability would give the paper a wider scope.**

A2.- A similar argument can be made as in the previous question. The method is validated for a particular numerical method and PDE, and three different ABLs, but could be applied to other methods easily as long as we can identify analogous parameters for optimization. Using N_ABL is natural for Cartesian-grid-based methods, but that parameter could be replaced with other analogous parameters that control the thickness of the ABL with respect to the minimum wavelength in the model. As a consequence we have added the following paragraph in the manuscript at the end of section 3.1:

"Moreover, It is important to highlight that the methodology for calibration of ABLs presented in this work is based upon three main components. Firstly, using representative models, secondly, establishing suitable metrics for absorption and finally, reducing the calibration to two parameters. We are not adding any assumptions regarding the underlaying PDEs used (linear acoustic waves, in our case). Similarly there are no assumptions tied to the numerical method (pseudospectral time-domain, in our case). Nevertheless two modifications are foreseen for broadening the applicability of the method. On one hand, in the case of using other physical models, we would need to modify Eq. (18) with an alternate energy proxy. On the other hand, in the case of using other numerical methods, we may need to replace N_ABL with an alternative parameter that is a measure of the thickness of the ABL with respect to the minimum wavelength. The actual results of the calibration,

of course, would be different for other PDEs and methods, but the calibration methodology is only expected to require the aforementioned, minor, modifications."

Finally, mention that these comments were also pointed out by the first referee, therefore we also include the modifications previously done into this final version. Other comments related to this issue can be found in the conclusions.

**Minor Comments**

**Q1.- Abstract, title of Sec. 2.4: Berenger's paper uses the term "Perfectly Matched Layer", not "Layers". Recommend sticking to the singular.**

A1.- Checked

**Q2.- Line 100: What does "finite in space and time" mean? Does this mean "bounded"?**

A2.- Yes, we replace the word.

**Q3.- Line 245: Gao et al. is repeated.**

A3.- Checked

**Q4.- Line 251: Replace "less" with "the least".**

A4.- Checked

**Q5.- Line 255: Eq. (18) is proportional to the (discrete) L^2 norm.**

A5.- It is specified into the text.

**Q6.- Figures 3 and 5: The vertical axis is label "energy" which is not precise since the quantity given by Eq. (19) does not have the units of energy. Recommend using the notation defined in Eq. (19) on vertical axis.**

A6.- Changed.

**Q7.- Figure 7: What are the units on the x and z axes?**

A7.- They represent the nodal mesh position at each coordinate. We prefer to keep it this way because we believe that the reader will appreciate that this figure is expressed in terms of numerical parameters, in this case, the number of nodes at each direction.

**Q8.- Line 478: "Such" should not be capitalized. Also, "put to the test" could be replaced with "tested".**

A8.- Changed.

**Q9.- Line 492: Need a space after "of".**

A9.- Checked.

Finally mention that we have included into the marked pdf all the changes due to the other reviewer comment's (including the afforementioned modifications in the major changes). We strongly believe that they improve the quality of the paper and are completely complementary to these changes.